# Trans-Endothelial Migration of Memory T Cells Is Impaired in Alemtuzumab-Treated Multiple Sclerosis Patients

**DOI:** 10.3390/jcm11216266

**Published:** 2022-10-24

**Authors:** Kristy Nguyen, Pierre Juillard, Simon Hawke, Georges E. Grau, Felix Marsh-Wakefield

**Affiliations:** 1Vascular Immunology Unit, School of Medical Sciences, Faculty of Medicine and Health, The University of Sydney, Sydney, NSW 2006, Australia; 2Central West Neurology and Neurosurgery, Orange, NSW 2800, Australia; 3Liver Injury and Cancer Program, Centenary Institute, Sydney, NSW 2050, Australia; 4Human Cancer and Viral Immunology Laboratory, The University of Sydney, Sydney, NSW 2006, Australia

**Keywords:** multiple sclerosis, alemtuzumab, trans-endothelial migration, blood–brain barrier, T cells

## Abstract

The breakdown of the blood–brain barrier (BBB) and the trans-endothelial migration of lymphocytes are central events in the development of multiple sclerosis (MS). Autoreactive T cells are major players in MS pathogenesis, which are rapidly depleted following alemtuzumab treatment. This modulation, in turn, inhibits CNS inflammation, but alemtuzumab’s effect on T cell migration into the CNS has been less studied. Human brain endothelial cells were stimulated with pro-inflammatory cytokines to mimic an inflamed BBB in vitro. Peripheral blood mononuclear cells from healthy controls, untreated or alemtuzumab-treated patients with relapsing-remitting MS (RRMS) were added to the BBB model to assess their transmigratory capacity. Here, the migration of CD4^+^ effector memory T (T^EM^) and CD8^+^ central memory T (T^CM^) cells across the BBB was impaired in alemtuzumab-treated patients. Naïve T (T^naïve^) cells were unable to migrate across all groups. CD38 was lowly expressed on CD8^+^ T^CM^ cells, particularly for RRMS patients, compared to CD8^+^ T^naïve^ cells. CD62L expression was lower on CD4^+^ T^EM^ cells than CD4^+^ T^naïve^ cells and decreased further in alemtuzumab-treated patients. These data suggest that repopulated memory T cells are phenotypically different from naïve T cells, which may affect their transmigration across the BBB in vitro.

## 1. Introduction

Alemtuzumab is a humanized monoclonal antibody that binds to CD52 expressed on T and B cells [1]. When administered to patients with relapsing-remitting MS (RRMS), alemtuzumab induces the rapid depletion of lymphocytes, followed by a slow recovery of precursor cells [2]. These quantitative changes slow the development of inflammatory demyelination, and one possible mechanism may be the inhibition of lymphocytes that migrate into the CNS. Many studies have implicated the role of autoreactive T cells in triggering disease onset using an experimental autoimmune encephalomyelitis (EAE) model [3,4]. Others have explored the transmigration of T cells in vitro and found greater CD4^+^ and CD8^+^ T cell passage across a stimulated blood–brain barrier (BBB) in MS patients compared to healthy controls [5,6]. These findings support that autoreactive T cells in the CNS can be detrimental, and suppressing these cells is crucial to dampening neuroinflammation.

Several studies have explored the immunomodulatory effects of alemtuzumab in vitro. Alemtuzumab has been reported to prompt cell maturation in a tolerogenic environment, with an increase in regulatory T cells [7,8]. An upregulation of inhibitory T cell receptors, such as PD-1 and LAG-3, and the reduction of autoreactive T cell clones, such as TCRβ CDR3, was also evident [7]. While alemtuzumab has been shown to replenish T cell responses [9], its remarkable efficacy in the clinical setting is well established. A phase II trial (CAMMS-223) and two phase III trials (CARE-MS I/II) revealed lasting suppression of disease activity and improved disability upon alemtuzumab treatment [10]. The possibility of adverse events, such as infusion reactions, infections, and secondary autoimmunity, has also been heavily considered with its use [11].

Many studies have investigated the effect of disease-modifying therapies on the BBB in vitro, including glatiramer acetate [12,13], interferon beta [12,13,14], fingolimod [15] and cladribine [16]. To date, only one murine study has explored alemtuzumab’s effect on cellular migration across the BBB in vitro. This study used a transwell assay to assess the migratory patterns of lymphocytes purified from anti-murine CD52-treated mice [17]. They found that CD4^+^ T cells retained the ability to migrate into the CNS, which was largely reflected in animals treated with LPS-induced inflammation in vivo [17]. Alemtuzumab also induced T cell depletion, where the remaining population was enriched in T cells with a regulatory phenotype [17]. Another study assessed BBB permeability in alemtuzumab-treated patients with RRMS [18]. Although increased breakdown of the BBB was found through two-year MRI outcomes, the characterisation of cells and their migratory capacity was not evaluated [18]. Further studies of alemtuzumab’s direct effect on the BBB are still yet to be investigated.

In addition to its depleting effects, alemtuzumab can induce qualitative changes in cell subsets that promote tolerance to the CNS [2,19]. Here, we investigate the migratory capacity of T cell subsets, including naïve T (T^naïve^) cells, effector memory (T^EM^) cells, central memory (T^CM^), and effector memory cells re-expressing CD45RA (T^EMRA^) in alemtuzumab-treated patients. Memory T cells mediate a faster response to antigen and confer long-lived protection compared to naïve T cells [20]. Strikingly, we found that CD4^+^ T^EM^ and CD8^+^ T^CM^ cells had an impaired ability to migrate in alemtuzumab-treated patients. CD38, CD49d, and CD62L expression differed between these cell subsets, which is hypothesised to influence migration across an inflamed BBB. Finally, the decreased level of circulating CD4^+^ T^EM^ and CD8^+^ T^CM^ cells following alemtuzumab treatment suggests these cells have recently repopulated. Together, these results provide deeper insights into disease pathogenesis and reveal a new therapeutic potential of alemtuzumab.

## 2. Materials and Methods

### 2.1. Study Participants

Patients with RRMS diagnosed using McDonald’s 2017 criteria [21] were recruited from Central West Neurology and Neurosurgery in Orange, NSW and from the Brain and Mind Centre at the University of Sydney. Subjects were eligible to participate in the study based on the inclusion and exclusion criteria shown in Appendix A. Ethical consent for the study was approved by the Research Integrity and Ethics Administration of the University of Sydney (Ethics Code 2018/708). After informed consent, blood samples were taken from healthy controls (n = 12), untreated RRMS patients (n = 6) and alemtuzumab-treated RRMS patients (n = 10). Age- and sex-matched healthy controls had samples taken from a single timepoint. Untreated MS patients had active disease defined by clinical and MRI activity within the previous 3 months. Blood from alemtuzumab-treated patients was collected after the first course in which immune cell reconstitution occurred (Lemtrada^®^, Sanofi-Genzyme, Cambridge, MA, USA; 5 consecutive days of 12 mg/day, then administered 12 months after for 3 consecutive days of 12 mg/day). All patients had variable Expanded Disability Status Scale scores (range 0–6.5). Data of the study participants are shown in Table 1.

### 2.2. PBMC Isolation from Whole Blood

Fresh whole blood was collected from healthy controls, untreated RRMS patients and alemtuzumab-treated RRMS patients using EDTA tubes kept at room temperature (15–20 °C). 30 mL of blood samples were obtained from the antecubital area, from either the median cubital, the cephalic or the basilic veins. Blood was diluted with equal amounts of phosphate-buffered saline (PBS) without calcium and magnesium (Sigma-Aldrich, St. Louis, MO, USA). Peripheral blood mononuclear cells (PBMC) were isolated from blood within 0–8 h after collection using Ficoll-PAQUE PLUS (GE Healthcare Pharmacia, Uppsala, Sweden) density gradient separation at 400× *g* for 30 min. After centrifugation, PBMCs were collected from the plasma/Ficoll-PAQUE interface and diluted with PBS-2%FBS. Centrifugation was repeated to remove any platelets or cellular debris, followed by resuspension of PBMCs in PBS-2%FBS. Cells were counted using a Countess II Automatic Cell Counter (Thermo Fisher Scientific, North Ryde, NSW, Australia), and the fresh cell sample was recorded as the concentration of cells in blood.

### 2.3. Transmigration Assay

Transmigration assays that were modified based on the Boyden^®^ chamber protocol were used as previously described [15] (Appendix A). In short, transwell inserts were used to create a two-compartment system consisting of an upper chamber and a lower chamber. Each cell culture insert had a polycarbonate membrane with 3 μm sized pores (Costar Pharma, Smithfield, NSW, Australia) to permit cell migration. On day 1, to emulate the physiological conditions of the BBB, transwell inserts were coated with a 3% collagen solution of rat tail collagen type I (Sigma-Aldrich, St. Louis, MO, USA) and sterile water (Baxter, Old Toongabbie, NSW, Australia). Each insert was incubated for at least 45 min at 37 °C to allow optimal cell attachment and then removed, leaving a translucent and adherent layer of collagen.

The human brain endothelial cell (HBEC) line, hCMEC/D3, was carefully seeded onto the collagen-coated inserts (300,000 HBECs/mL, 450,000 HBECs per transwell) then laid for 20 min to allow firm adhesion. Subsequently, 2.6 mL of warmed 1% antibiotic antimycotic (AA) solution (Sigma-Aldrich, St. Louis, MO, USA) in complete EBM-2 medium (Lonza, Basel, Switzerland) supplemented with 5% FBS (Sigma-Aldrich), 5 μg/mL ascorbic acid (Sigma-Aldrich), 1.4 μmol/L hydrocortisone (Sigma-Aldrich), CDLC (1:100 dilution; Life technologies, Carlsbad, CA, USA), 10 mmol/L HEPES (Sigma-Aldrich) and 1 ng/mL β-FGF (Sigma-Aldrich) was added to the lower compartment of the cell culture inserts and incubated at 37 °C and 5% CO_2_ overnight. On days 2 and 3, spent medium in the upper and lower chambers was removed and each insert was transferred to a new 6-well plate. Warmed 1% AA in complete EBM-2 medium was added to both the wells and transwell inserts (2.6 mL and 1.5 mL, respectively). Transwell inserts were left to incubate overnight at 37 °C and 5% CO_2_.

On day 4, to replicate the pathological conditions of MS, transwell inserts were refreshed with warmed 1% AA in complete EBM-2 medium supplemented with pro-inflammatory cytokines, TNF (5 ng/mL, PeproTech, Rocky Hill, NJ, USA) and IFN-γ (10 ng/mL, PeproTech) then incubated at 37 °C and 5% CO_2_. Activation of the HBEC monolayer simulated an inflamed BBB and allowed immune cell migration.

On day 5, a morphological assessment of the HBEC monolayer was performed with Wheat Germ Agglutinin (WGA) Alexa Fluor™ 555 conjugate (Invitrogen, Thermo Fisher Scientific, NJ, USA) at least 3 h before using the transmigration assay. WGA is a fluorescent tool that selectively binds to N-acetylglucosamine and N-acetylneuraminic (sialic) acid residues, allowing whole staining of the HBEC monolayer. Each insert was transferred to a new 6-well plate filled with warmed WGA solution (5 µg/mL; 2.5 mL into well and 1 mL into transwell insert) and incubated at room temperature in the dark for 10 min. Inserts were then washed twice with PBS and transferred to a new 6-well plate filled with warmed 1% AA in complete EBM-2 medium (5 µg/mL; 2.5 mL into well and 1 mL into transwell insert). Using fluorescence microscopy, the HBEC monolayer was checked for the absence of holes or accumulation of cell layers to ensure optimal confluency. The inserts were washed twice with PBS followed by 1.5 mL of warmed 1% AA in complete EBM-2 medium. After confirming the integrity of the HBEC monolayer, each insert was transferred to a new 6-well plate filled with 2.6 mL of warmed 3% FBS in RPMI 1640 medium (Sigma-Aldrich, St. Louis, MO, USA). 2.34 × 10^6^ of isolated PBMCs (1.56 × 10^6^/mL) resuspended in 3% FBS/RPMI was then added to the cell culture inserts and incubated at 37 °C and 5% CO_2_ for 14–18 h.

After the transmigration, non-migrated cells in the upper chamber of the transwell were collected without disrupting the endothelial monolayer. Cell culture inserts were removed and migrated cells in the lower chamber were harvested. Non-migrated and migrated cell samples were counted using a Countess II Automatic Cell Counter (Thermo Fisher Scientific, North Ryde, NSW, Australia), and the percentage of viability was calculated using Trypan blue staining prior to flow cytometric analysis.

### 2.4. Spectral Flow Cytometry

Flow cytometric analysis was performed on the same day as the isolation of PBMCs (fresh cell sample) or PBMC collection from the transmigration assay (non-migrated and migrated cell sample). PBMCs were centrifuged for 5 min at 500× *g* at room temperature, after which each pellet was resuspended in FACS buffer (PBS supplemented with 0.5% BSA Sigma Chemical Co., St Louis, MO, USA) and 2 mM EDTA (AMRESCO Inc, Solon, OH, USA). PBMCs were then incubated for 20 min at 4 °C in a staining mix containing FACS buffer and fluorescently conjugated antibodies (Appendix A), washed twice and fixed in 4% paraformaldehyde (PFA) for 20 min at room temperature. Cell samples were quantified using the Aurora flow cytometer (Cytek^®^, Fremont, CA, USA) and then analysed using FlowJo software v10.7 (BD, Ashland, Oregon, USA).

### 2.5. Gating Strategy

Forward-scatter-height (FSC-H) and time were first gated on. Total PBMCs were gated based on their forward-scatter-area (FSC-A) and side-scatter-area (SSC-A) to remove any cellular debris and dead cells. Single cells were further gated using FSC-A and FSC-H to exclude doublets and triplets.

T cells were identified as CD3^+^CD19^−^ (Appendix A). T cells were then gated to identify CD4^+^ and CD8^high/low^ T cells. CD4^+^, CD8^high^ and CD8^low^ T cells were further gated to identify CD20^+^ T cells and CD56^+^ NKT cells. The delineation of CD8^+^ T cells into CD8 high-expressing (CD8^high^) and CD8 low-expressing (CD8^low^) T cells has been previously studied in MS patients [22]. Remaining T cells (CD20^−^CD56^−^) were subdivided based on CD197 and CD45RA expression. The distinct subsets of T cells correspond to terminally differentiated effector memory cells re-expressing CD45RA (T^EMRA^, CD45RA^+^CD197^−^), naïve (T^naïve^, CD45RA^+^CD197^+^), central memory (T^CM^, CD45RA^−^CD197^+^) and effector memory (T^EM^, CD45RA^−^CD197^−^) T cells (Appendix A). These subsets were gated to calculate their proportions and then multiplied by their cell counts to obtain the total cell numbers.

### 2.6. Statistical Analysis

Statistical analyses were performed with Prism 9.0.2 GraphPad software (San Diego, California, USA). A Kruskal–Wallis nonparametric one-way ANOVA with a Dunn’s multiple comparisons test was used to compare values across test groups. A Wilcoxon matched-pairs signed-rank test was used to compare paired migrated and non-migrated cell samples in test groups. For log2 fold changes, a one sample nonparametric Wilcoxon signed-rank test with Pratt method was used to compare the median of test groups to a hypothetical value of 0. Here, we determined the ratio of migrated to non-migrated cells then provided the MFI as representation of the data. Log2 is indicated by a 4-fold increase in MFI, with higher levels showing more expression of a given marker. Values denoted by *p* ≤ 0.1 are shown for all graphs.

## 3. Results

### 3.1. CD4^+^ T^EM^ and CD8^+^ T^CM^ Cells from Alemtuzumab-Treated RRMS Patients Have Reduced Trans-Endothelial Migration

For each transmigration assay, cells that were migrating (cells found in the lower chamber) and cells that were non-migrating (cells remaining in the upper chamber) were characterised. The migrated and non-migrated cell samples were measured as total cell numbers and presented as paired individual values. In healthy controls and untreated RRMS patients, CD4^+^ T^EM^ cells freely transmigrated (Figure 1A(i)). Conversely, the majority of CD4^+^ T^EM^ cells in the alemtuzumab-treated group did not migrate. There was no difference in the migratory ability of CD8^high^ T^CM^ cells between healthy controls and untreated RRMS patients (Figure 1B(i)). Alemtuzumab-treated patients had fewer CD8^high^ and CD8^low^ T^CM^ cells migrating across the stimulated BBB (Figure 1B–C(i)). Unlike their counterpart, the naïve phenotypes did not show active migration across the BBB for all groups (Figure 1A–C(ii)). The transmigration of other CD4^+^ and CD8^+^ T cell subsets varied, including complete migration of CD8^+^ T^EMRA^ cells (Appendix A).

### 3.2. CD38 and CD62L Expression on CD4^+^ T^EM^ and CD8^+^ T^CM^ Cells Is Altered following Alemtuzumab

To determine how alemtuzumab may influence T cell migration, the expression of CD38, CD49d, and CD62L were explored on freshly isolated PBMCs. The mean fluorescence intensity (MFI) for these markers was quantified on CD4^+^ T^EM^ and CD8^high/low^ T^CM^ cells, as well as their naïve phenotypes. For all study groups, CD4^+^ T^naïve^ and CD8^high/low^ T^naïve^ cells displayed a higher expression of CD38 compared to CD4^+^ T^EM^ cells and CD8^high/low^ T^CM^ cells, respectively (Figure 2A(i–iii)). Notably, alemtuzumab-treated patients showed higher CD38 expression for CD4^+^ T^EM^ cells compared to untreated RRMS patients (Figure 2A(i)). Memory T cell subsets displayed higher CD49d expression relative to their naïve phenotype (Figure 2B(i–iii)). Like CD38, CD62L was more highly expressed on T^naïve^ cells compared to memory phenotypes (Figure 2C(i–iii)). Lower CD62L expression, particularly on CD4^+^ T^EM^ cells, was shown in alemtuzumab-treated patients compared to other groups (Figure 2C(i)). Marker density in non-migrated and migrated memory T cells are detailed in Appendix A.

### 3.3. Alemtuzumab Significantly Depletes Circulating CD4^+^ T^EM^ and CD8^+^ T^CM^ Cell Numbers

To determine whether differences in transmigrated cells were due to depletion in the periphery, the number of circulating cells from freshly isolated PBMCs was calculated. The total number of circulating CD4^+^ T^EM^ and CD8^high/low^ T^CM^ cells, as well as their naïve phenotypes, were reduced after alemtuzumab treatment compared to healthy controls and untreated RRMS patients (Figure 3A,B(i–iii)). The total number of fresh PBMCs and major lymphocyte populations are shown in Appendix A. This is similarly displayed for other CD4^+^ and CD8^+^ T cell subsets in Appendix A.

## 4. Discussion

In the present study, the migratory capacity of CD4^+^ T^EM^ and CD8^+^ T^CM^ cells across the BBB in vitro was examined. Modulation of CD8^+^ T^CM^ cells has previously been explored in MS therapy. Upon treatment with interferon beta, CD8^+^ T^CM^ cell proportions increase due to a shift in naïve regulatory T cells to a ‘T^CM^-like’ regulatory T cell population [23]. In contrast, fingolimod reduces the proportion of CD8^+^ T^CM^ cells [15]. CD8^+^ T^CM^ cells have a greater proliferative and self-renewal capacity, particularly in response to pleiotropic cytokines such as IL-7, IL-12, and IL-15 [24,25]. Remarkably, alemtuzumab treatment was shown to reduce Th1 and Th17 responses, leading to a marked decrease in IL-12 and IL-15 [2,8,26], which are known to influence the migration of CD8^+^ T^EM^ and T^EMRA^ cells in vivo [27]. Unlike T^CM^ cells, T^naïve^ cells typically remain at lower frequencies and require direct contact with their cognate antigen to enter the CNS [3]. These antigen-inexperienced cells also lack homing and chemokine receptors to migrate into peripheral tissues [20]. By reducing inflammation, alemtuzumab allows T^naïve^ cells to slowly reconstitute and preserve immunocompetence for newly matured cells.

T cells undergo various stages of differentiation and activation in physiological conditions [28,29]. T^naïve^ cells express abundant levels of CD38 compared to mature T cells [30,31]. Higher expression of this marker in alemtuzumab-treated patients may be due to repleted precursor cells that modulate cell recruitment, cell activation, and cytokine and chemokine release, without inducing an inflammatory cascade [32]. While resting T cells can lack expression of CD38, it can be upregulated by cytokine and MHC activation [33]. Lower expression of CD38 on T^EM^ and T^CM^ cells may explain their altered migration across the BBB, especially for alemtuzumab-treated patients. This could be due to resting memory T cells from the bone marrow that emigrate to the periphery, with similar kinetics to T^naïve^ cells in traversing the BBB and no pathological features like autoreactive cells prior to alemtuzumab treatment.

In contrast to CD38, CD49d was highly expressed on T^EM^ and T^CM^ cells compared to their naïve phenotype. CD49d expression is higher in untreated RRMS patients due to the activation of the BBB in response to pro-inflammatory stimuli [34]. CD49d upregulation by an inflamed BBB can mediate the tethering and rolling of leukocytes upon binding to VCAM-1 on endothelial cells, allowing migration into the CNS [35]. CD49d has an inactive conformation on resting cells [36], so its low expression on T^naïve^ cells may explain their inability to migrate. Interestingly, most CD4^+^ T^EM^ and CD8^+^ T^CM^ cells in alemtuzumab-treated patients displayed high expression of CD49d despite limited migration across the BBB. Natalizumab inhibits CD49d expression on PBMCs, which interferes with CNS migration [37]. However, altered migration of CD4^+^ T^EM^ and CD8^+^ T^CM^ cells in alemtuzumab-treated patients suggests that these cells may use a CD49d-independent pathway to migrate across the BBB.

CD62L undergoes proteolytic shedding in the lymph nodes during T cell activation and re-enters into peripheral circulation to exert effector functions [38]. This would result in a lower density of CD62L remaining on the T cell surface. T^naïve^ and T^CM^ cells highly express CD62L, unlike T^EM^ cells which display lower levels [22]. This was reflected in this study, as T^EM^ cells have relatively lower expression of CD62L compared to T^CM^ cells, particularly for CD4^+^ T cells. Interestingly, CD62L expression on CD8^high/low^ T^CM^ cells was lower than CD4^+^ T^EM^ cells in the alemtuzumab-treated patients. This could indicate that CD8^+^ T^CM^ cells are slowly reconstituting due to their capacity for self-renewal, however, possess a lower migratory capacity due to the lack of antigen encounter and activation [39]. Hence, the downregulation of CD62L in memory and naïve phenotypes could suggest repopulation from the lymph nodes to aid immune surveillance.

Notably, alemtuzumab reduced levels of CD4^+^ T^EM^ and CD8^+^ T^CM^ cells, and their naïve phenotypes, compared to other test groups. CD4^+^ and CD8^+^ T^naïve^ cells are more susceptible to alemtuzumab depletion and undergo slow repopulation than memory T cells [40,41]. Following lymphopenia, fewer T cells that traverse the BBB can mitigate disease activity, and may also allow T cell recovery driven by homeostatic proliferation [7]. While CD38 can be upregulated during inflammation, recently matured cells that highly express this marker in alemtuzumab-treated patients may have proliferated from non-depleted memory T cells. Moreover, high expression of CD49d on T^naïve^ cells following alemtuzumab supports the notion of a CD49d-independent mechanism. This may explain why memory T cells also express high levels of CD49d but are unable to migrate across the stimulated BBB. Despite low expression of CD62L across all groups, alemtuzumab-treated patients display lower levels of this marker due to recent emigrants from the lymph nodes that survey the periphery.

One limitation was the small sample size of the untreated MS cohort (n = 6) and the lack of paired timepoints. The best neurological practice is to treat MS patients without delay to avoid the risk of irreversible damage. However, future studies would benefit from a larger cohort. Another limitation was the technical set-up of the in vitro BBB model to reflect the in situ environment. In this study, a transmigration assay was used to examine lymphocyte migration across the BBB as shown by other studies [15,42,43]. The brain microenvironment is difficult to replicate, particularly in the context of inflammation. BBB cell types and structures, such as astrocytes, pericytes and the glia limitans, are other regulators of the CNS that should be considered [44]. These additional cell types may be closely linked with CD38, CD49d, and CD62L to allow migration across the BBB, predicting a potential role in MS pathogenesis and during alemtuzumab treatment. Although a technical caveat, the preparation of a co-culture with astrocytes and pericytes may not be feasible when collecting and testing fresh blood samples. A more complete in vitro BBB model that incorporates more cells may be considered in future studies.

## 5. Conclusions

In summary, we investigated the migratory capacity of T cell subsets in alemtuzumab-treated and treatment naïve RRMS patients and found that in alemtuzumab-treated patients, CD4^+^ T^EM^ cells and CD8^+^ T^CM^ cells had defective migration in our BBB model. Newly matured cells that have transitioned from their naïve phenotypes may be attributed to the varying levels of activation or adhesion molecules that differ in MS conditions. We established that alemtuzumab has long-term immunomodulatory effects that impede processes at the molecular level. Data produced from this study extensively focused on the phenotypic profiles of T cell subsets. This could reveal a new mechanism of action for alemtuzumab regarding BBB migration and may enhance our understanding of MS pathophysiology.

## Figures and Tables

**Figure 1 jcm-11-06266-f001:**
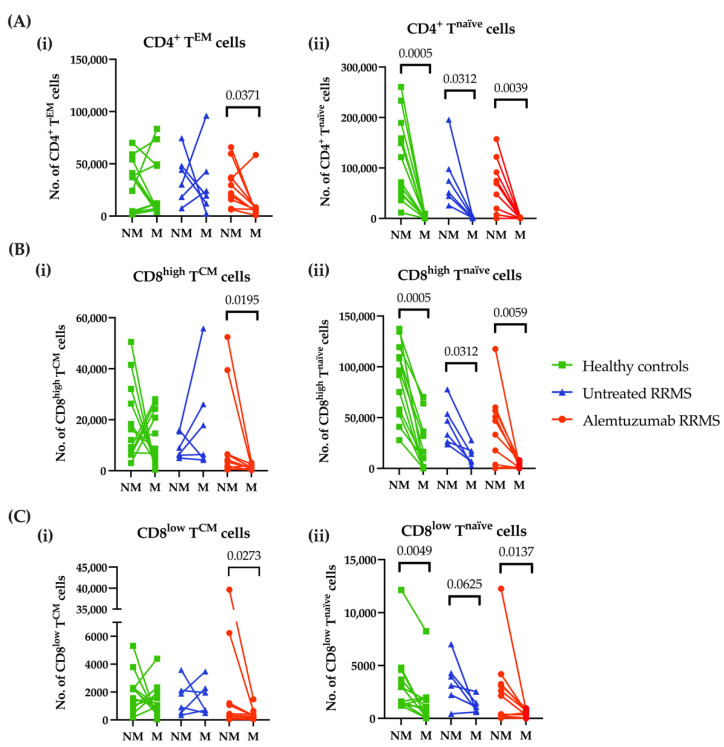
**Alemtuzumab decreased the migratory capacity of CD4^+^ T^EM^ and CD8^+^ T^CM^ cells across the BBB in vitro.** PBMCs were isolated from healthy controls (n = 12, green squares), untreated RRMS (n = 6, blue triangles) and alemtuzumab-treated RRMS patients (n = 10, red circles). After stimulation of the HBEC monolayer, PBMCs were added to the transwell assay and were left overnight to migrate. Non-migrated and migrated cells were harvested from the BBB model and phenotyped for analysis as total cell numbers. (**A**) The number of (**i**) CD4^+^ T^EM^ cells and (**ii**) CD4^+^ T^naïve^ cells. (**B**) The number of (**i**) CD8^high^ T^CM^ cells and (**ii**) The CD8^high^ T^naïve^ cells. (**C**) The number of (**i**) CD8^low^ T^CM^ cells and (**ii**) CD8^low^ T^naïve^ cells. Wilcoxon matched-pairs signed-rank test. *p* ≤ 0.1 are shown. NM; non-migrating, M; migrating; T^CM^, central memory T cells; T^EM^, effector memory T cells; T^naïve^, naïve T cells; RRMS, relapsing-remitting multiple sclerosis.

**Figure 2 jcm-11-06266-f002:**
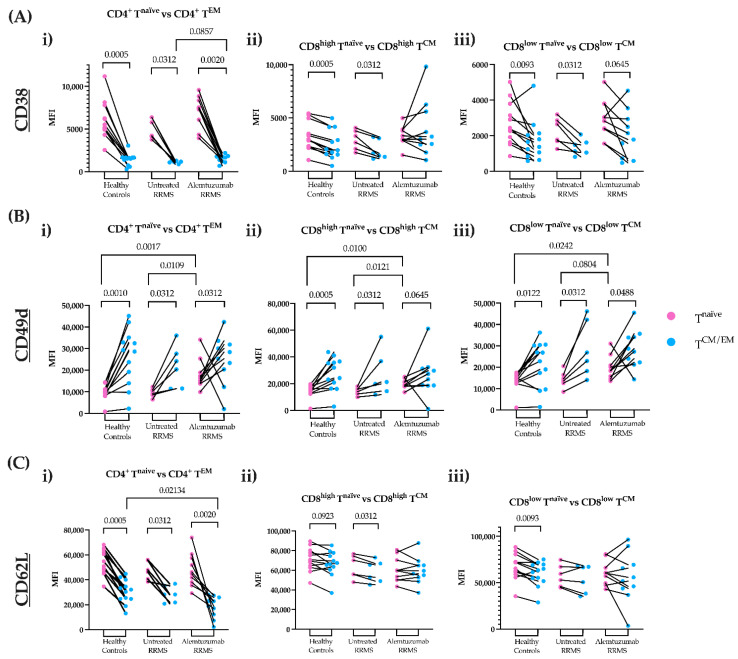
**The mean fluorescence intensity (MFI) of CD38, CD49d and CD62L on fresh CD4^+^ T^EM^ and CD8^+^ T^CM^ cells compared to their naïve phenotypes.** The expression of cell surface markers was calculated in healthy controls (n = 12), untreated RRMS (n = 6) and alemtuzumab-treated RRMS patients (n = 10) for naïve (pink) and memory (blue) T cells. (**A**) CD38 expression on (**i**) CD4^+^ T^EM^ cells versus CD4^+^ T^naïve^ cells (**ii**) CD8^high^ T^CM^ cells versus CD8^high^ T^naïve^ cells (**iii**) CD8^low^ T^CM^ cells versus CD8^low^ T^naïve^ cells. (**B**) CD49d expression on (**i**) CD4^+^ T^EM^ cells versus CD4^+^ T^naïve^ cells (**ii**) CD8^high^ T^CM^ cells versus CD8^high^ T^naïve^ cells (**iii**) CD8^low^ T^CM^ cells versus CD8^low^ T^naïve^ cells. (**C**) CD62L expression on (**i**) CD4^+^ T^EM^ cells versus CD4^+^ T^naïve^ cells (**ii**) CD8^high^ T^CM^ cells versus CD8^high^ T^naïve^ cells (**iii**) CD8^low^ T^CM^ cells versus CD8^low^ T^naïve^ cells. Wilcoxon matched-paired signed-rank test compared paired naïve and memory T cells. A Kruskal–Wallis with Dunn’s multiple comparisons test was done to compare between groups. *p* ≤ 0.1 are shown. RRMS, relapsing-remitting multiple sclerosis; T^CM^, central memory T cells; T^EM^, effector memory T cells; T^naïve^, naïve T cells.

**Figure 3 jcm-11-06266-f003:**
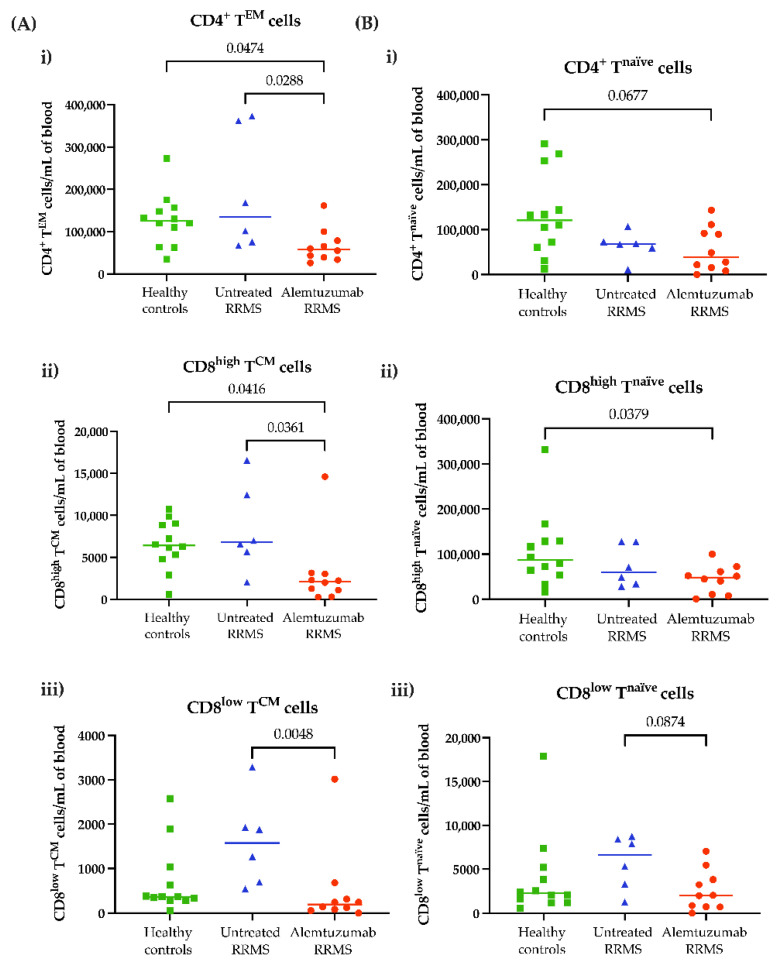
**Alemtuzumab depletes circulating T cell subsets.** Blood was sampled from healthy controls (n = 12, green squares), untreated RRMS (n = 6, blue triangles) and alemtuzumab-treated RRMS patients (n = 10, red circles). Upon sample collection, fresh PBMCs were immediately isolated and phenotyped. Each T cell subset was gated on to calculate the proportion of cells multiplied by the cell count and shown as cells per mL of blood. (**A**) The absolute number of (**i**) CD4^+^ T^EM^ cells, (**ii**) CD8^high^ T^CM^ cells and (**iii**) CD8^low^ T^CM^ cells. (**B**) The absolute number of **(i)** CD4^+^ T^naïve^ cells, (**ii**) CD8^high^ T^naïve^ cells and (**iii**) CD8^low^ T^naïve^ cells. Kruskal–Wallis with Dunn’s multiple comparisons test. *p* ≤ 0.1 are shown. T^CM^, central memory T cells; T^EM^, effector memory T cells; T^naïve^, naïve T cells. RRMS, relapsing-remitting multiple sclerosis.

**Table 1 jcm-11-06266-t001:** Demographic and clinical characteristics of study participants.

MS ID	Sex (62.5% F)	Age * (Median = 42.6)	Disease Duration (Years) *	Months Since First Dose		Previous Treatment	Months Since Last Treatment Prior to Alemtuzumab	Active MS **
				Prior	Post			
MS01	F	36	9.3	-	21	Azathioprine & intravenous immunoglobulin	1	Yes
MS02	F	40.2	6.9	-	11	Dimethyl fumarate	3	Yes
MS03	F	38.9	5.9	-	12	Fingolimod	2	Yes
MS04	M	55.2	3.1	-	19	Fingolimod	3	
MS05	M	46.4	8.8	-	8	Fingolimod	2	Yes
MS06	M	43.1	23.3	-	19	Natalizumab	1	Yes
MS07	M	57.5	19.1	-	54	Interferon beta & natalizumab	3	Yes
MS08	F	44.9	5.8	-	50	Dimethyl fumarate	2	Yes
MS09	F	37.7	10.1	-	54	Fingolimod & natalizumab	3	Yes
MS10	F	42.1	12.1	-	37	Interferon beta, fingolimod & natalizumab	2	Yes
MS11	F	52	0.1	✓	-	Mavenclad	-	Yes
MS12	F	28.3	0.3	✓	-	None	-	Yes
MS13	F	37.9	6.3	✓	-	None	-	Yes
MS14	M	29.6	0.1	✓	-	None	-	Yes
MS15	F	61.7	24	✓	-	None	-	Yes
MS16	M	47	0.1	✓	-	None	-	Benign MS
**Healthy Control ID**				**Sex (66% F)**			**Age * (Median = 38)**	
HC01				F			24	
HC02				M			26	
HC03				M			22	
HC04				F			35	
HC05				F			30	
HC06				F			41	
HC07				M			43	
HC08				F			56	
HC09				F			28	
HC10				F			42	
HC11				M			43	
HC12				F			41	

F, female; M, male; MS, multiple sclerosis; Post, 8–54 months after the first dose; Prior, MS before alemtuzumab. * Age/disease duration when first sample was taken. ** Active MS defined as new T2 or T1 Gad-enhancing lesions in the 8 months prior to starting alemtuzumab treatment.

## Data Availability

The data presented in this study are available from Sanofi. Restrictions apply to the availability of these data, which were used under license for this study. Data are available upon request to the authors with the permission of Sanofi.

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
