# Peer review of "Trans-Endothelial Migration of Memory T Cells Is Impaired in Alemtuzumab-Treated Multiple Sclerosis Patients"

_jcm, 2022, doi:10.3390/jcm11216266_

Round 1

Reviewer 1 Report

Reviewer’s comments

In this study, using in vitro model of BBB, the authors have examined the migratory capacity of CD4+ TEM and CD8+ TCM cells in healthy control, untreated and alemtuzumab-treated, relapsing-remitting multiple sclerosis patients. They found that alemtuzumab-treated patients, CD4+ TEM cells and CD8+ TCM cells had defective migration in BBB model.

This manuscript was well-written and would be helpful to understand the mechanism of action of alemtuzumab in BBB migration of T cells and MS pathophysiology which in turn raising potential role of alemtuzumab in therapeutic application.

General comments

- Are there any side-effects of alemtuzumab?

- Could use alemtuzumab for long-term?

- Can alemtuzumab directly act on BBB permeability?

- By suppression of T cell migration to brain across BBB, it could be reduced inflammation, but how do you think the effects of lymphopenia?

Specific comments

-          What are the inclusion and exclusion criteria for participants?

-          How many ml of blood was drawn and from which vein?

-          How about total and differential count of participants?

-          Are there any literatures for T cell migration to brain inducing neuroinflammation and cognitive deficits?

-          How about the other inflammatory markers in treated and untreated patients.

-          In discussion, although this study is in vitro study, the authors should mention when T cell migration into brain through BBB, how about the neuro-glia (microglia and astrocyte) -mast cell interaction or crosstalk to rescue or enhance the MS?

Author Response

Are there any side-effects of alemtuzumab?

  • Added to the introduction in lines 43-44.

Could use alemtuzumab for long-term?

  • Added to the introduction in lines 39-43.

Can alemtuzumab directly act on BBB permeability?

  • Added to the introduction lines 66-67.

By suppression of T cell migration to brain across the BBB, it could be reduced inflammation, but how do you think the effects of lymphopenia?

  • Added to the discussion in lines 455-457.

What are the inclusion and exclusion criteria for participants?

  • We added an extra supplementary figure (Table S1).

How many mL of blood was drawn and from which vein?

  • We added an extra sentence to the methods in lines 98-99.

How about total and differential count of participants?

  • We added an extra supplementary figure (Figure S6).

Are there any literatures for T cell migration to brain inducing neuroinflammation and cognitive deficits? 

  • We added sentences to the introduction in lines 35-42.

How about the other inflammatory markers in treated and untreated patients.

  • There are other inflammatory markers that are important, however, we chose these specific markers because they play a major role in T cell migration
  • Our study had a more focused approach so analysing other markers was outside the scope of this study

In discussion, although this study is in vitro study, the authors should mention when T cell migration into brain through BBB, how about the neuro-glia (microglia and astrocyte) -mast cell interaction or crosstalk to rescue or enhance the MS?

  • We addressed this at the end of the discussion in lines 492-495.

Reviewer 2 Report

In this publication the authors aimed to investigate the impact of alemtuzumab on the trans-endothelial migration of T cells in alemtuzumab-treated patients suffering of multiple sclerosis. 

The authors used in vitro experiments using an artificial blood brain barrier and conclude that trans-endothelial migration of memory T cells is impaired in alemtuzumab-treated MS patients. 

Although the authors present their results I suggest some minor corrections:

Figure 2 is overlapping with the text.

Provide higher resolution of the images. 

Please provide a larger sample size in particular of the group of untreated MS patients and homogenize the investigated timepoints.

To further convince the readers of the experimental design please provide videos of the transmigration process. 

Author Response

Minor corrections

  • Figure 2 overlapping with text
    • Made sure this was not a problem in the latest version
  • Provide higher resolution of images
    • We used higher resolution images for embedment in the text
  • Please provide a larger sample size in particular of the group of untreated MS patients and homogenize the investigated timepoints
    • It was not possible to recruit more patients during the COVID-19 pandemic, so this was beyond the scope of the study
  • To further convince the readers of the experimental design please provide videos of the transmigration process
    • Our protocol mentions that we stained the membrane to confirm the integrity of the monolayer, and allow for proper cell migration across the stimulated BBB
    • Videos have been previously published to show the experimental design
      • https://www.jove.com/v/5644/the-transwell-migration-assay
      • https://www.jove.com/v/51046/in-vitro-cell-migration-and-invasion-assays

Additional changes

  • We made some minor grammatical/spelling/punctuation errors.
  • We added a study that similarly categorised CD8+ T cells into a CD8high and CD8low population (lines 205-206).
  • We clarified the use of Log2 Fold Change (Log2FC) in lines 218-220.
  • We added more references throughout the paper.

Round 2

Reviewer 2 Report

-